# Isolation and Genomic Analysis of 3-Chlorobenzoate-Degrading Bacteria from Soil

**DOI:** 10.3390/microorganisms11071684

**Published:** 2023-06-28

**Authors:** Ifat Ara, Ryota Moriuchi, Hideo Dohra, Kazuhide Kimbara, Naoto Ogawa, Masaki Shintani

**Affiliations:** 1Department of Environment and Energy Systems, Graduate School of Science and Technology, Shizuoka University, 3-5-1 Johoku, Naka-ku, Hamamatsu 432-8561, Japan; ara.ifat.18@shizuoka.ac.jp (I.A.); kimbara.kazuhide@shizuoka.ac.jp (K.K.); 2Functional Genomics Section, Shizuoka Instrumental Analysis Center, Shizuoka University, 836 Oh-ya, Suruga-ku, Shizuoka City 422-8529, Japan; moriuchi.ryota@shizuoka.ac.jp (R.M.); dora.hideo@shizuoka.ac.jp (H.D.); 3Department of Science, Graduate School of Integrated Science and Technology, Shizuoka University, 836 Oh-ya, Suruga-ku, Shizuoka City 422-8529, Japan; 4Department of Agriculture, Graduate School of Integrated Science and Technology, Shizuoka University, 836 Oh-ya, Suruga-ku, Shizuoka City 422-8529, Japan; ogawa.naoto@shizuoka.ac.jp; 5Department of Engineering, Graduate School of Integrated Science and Technology, Shizuoka University, 3-5-1 Johoku, Naka-ku, Hamamatsu 432-8561, Japan; 6Japan Collection of Microorganisms, RIKEN BioResource Research Center, 3-1-1 Koyadai, Tsukuba 305-0074, Japan; 7Research Institute of Green Science and Technology, Shizuoka University, 3-5-1 Johoku, Naka-ku, Hamamatsu 432-8561, Japan

**Keywords:** 3-chlorobenzoate, degradation, *Caballeronia*, *Paraburkholderia*, *Cupriavidus*, chlorocatechol, hydroxybenzoate, protocatechuate, gentisate

## Abstract

The compound 3-chlorobenzoate (3-CBA) is a hazardous industrial waste product that can harm human health and the environment. This study investigates the physiological and genetic potential for 3-chlorobenzoate (3-CBA) degradation. Six 3-CBA Gram-negative degraders with different degradation properties belonging to the genera *Caballeronia*, *Paraburkholderia* and *Cupriavidus* were isolated from the soil. The representative strains *Caballeronia* 19CS4-2 and *Paraburkholderia* 19CS9-1 showed higher maximum specific growth rates (*µmax*, h^−1^) than *Cupriavidus* 19C6 and degraded 5 mM 3-CBA within 20–28 h. Two degradation products, chloro-*cis*,*cis*-muconate and maleylacetate, were detected in all isolates using high-performance liquid chromatography and mass spectrometry. Genomic analyses revealed the presence of *cbe* and *tfd* gene clusters in strains 19CS4-2 and 19CS9-1, indicating that they probably metabolized the 3-CBA via the chlorocatechol *ortho*-cleavage pathway. Strain 19C6 possessed *cbe* genes, but not *tfd* genes, suggesting it might have a different chlorocatechol degradation pathway. Putative genes for the metabolism of 3-hydroxybenzoate via gentisate were found only in 19C6, which utilized the compound as a sole carbon source. 19C6 exhibited distinct characteristics from strains 19CS4-2 and 19CS9-1. The results confirm that bacteria can degrade 3-CBA and improve our understanding of how they contribute to environmental 3-CBA biodegradation.

## 1. Introduction

Halogenated chemicals continuously enter the environment owing to the increasing world population and industrial pollution and persist for decades in the environment [1,2,3]. The U.S. Environmental Protection Agency has listed these compounds as “high priority” pollutants under the Toxic Substances Control Act (TSCA) [4]. Several biological treatment technologies have shown promise in the removal of halogenated compounds. These technologies include bioscrubbers (e.g., trichloroethylene [TCE] and 1,2-dichloroethylene [1,2-DCE]) [5,6,7], biotrickling filters (e.g., o-chlorotoluene and m-dichlorobenzene) [8,9], biofilters (e.g., chlorobenzene and TCE) [10,11] and biodegradation.

Biodegradation is one of the most effective processes for removing toxic compounds from contaminated environments. This process allows bacteria to transform toxic compounds into nontoxic compounds using the compound as a carbon and energy source [12,13,14,15,16].

3-Chlorobenzoate (3-CBA) is a halogenated compound used as a raw material for producing fertilizers and for several applications in industry. In addition, 3-CBA can be produced as a dead-end product when polychlorinated biphenyls (PCB) are metabolized by microorganisms and 3-CBA degradation is important for complete PCB degradation [17]. The degradation of 3-CBA is carried out mainly through microbial metabolism. The mechanisms responsible for the ability of microorganisms to degrade 3-CBA have been a subject of considerable interest in biodegradation research [18,19]. 3-CBA has been widely used as a model compound in research on microbial degradation of aromatic chlorine compounds because of its relatively low toxicity compared with other chlorinated aromatic compounds such as PCBs, making it a suitable compound for experimental studies [20].

Table 1 lists a variety of pathways for 3-CBA degradation by different bacteria. 3-CBA degradation via (chloro-)catechol (CC) is the most common pathway, initiated by dioxygenase and converts 3-CBA to CC, which is usually channeled into the central metabolism by the CC *ortho*-cleavage pathway [18,21,22,23]. Only a few bacteria can use the meta-cleavage pathway to degrade CC [24,25,26]. CC can also be used as an important central intermediate in the degradation of various chloroaromatic compounds [27,28]. CC dioxygenase is an important enzyme for CC degradation and this, combined with its involvement in the ortho pathway, plays a crucial role in the microbial degradation of chloroaromatic compounds [21,29].

Alternatively, based on analyses of the intermediate compounds of 3-CBA degradation, it has been predicted that 3-CBA can also be degraded via 4-hydroxybenzoate (4-HB) or 3-hydroxybenzoate (3-HB) [30,31,32,33,34]. These compounds are converted to protocatechuate (PC) or gentisate (GS) and, therefore, these pathways are known as the PC and GS pathways, respectively [30,31,32,33,34]. The initial enzyme responsible for the conversion of 3-CBA to 4-HB or 3-HB has not yet been identified [30,31,32,33,34].

This study investigates the physiological and genetic potential of bacteria for 3-CBA degradation. Six 3-CBA-degrading bacteria of different genera were isolated from soil samples. Genome-wide analyses of the isolates were performed, including the identification of the isolates and putative genes involved in 3-CBA degradation.
microorganisms-11-01684-t001_Table 1Table 1Previously isolated bacterial strains and their pathways for 3-chlorobenzoate degradation.Bacterial Strain3-CBA Degradation PathwayReference*Caballeronia* sp. NK8CC[23]*Cupriavidus necator* NH9CC[16]*Pseudomonas* sp. B13CC[35]*Pseudomonas* sp. 3-CBACC[17]*Alcaligenes* sp. L6GS and PC[32]*Bacillus* sp. OS13PC[30]*Pseudomonas*GS[31]Abbreviations: 3-CBA, 3-chlorobenzoate; CC, chlorocatechol; GS, gentisate; PC, protocatechuate.


## 2. Materials and Methods

### 2.1. Growth Medium and Culture Conditions

The isolates in this study were cultured at 30 °C on Luria-broth (LB) [36] or carbon-free basal salt medium (BSM) [16] supplemented with 5 mM 3-CBA as the sole carbon and energy source and plates containing 1.7% agar at 30 °C.

### 2.2. Isolation of 3-Chlorobenzoate-Degrading Bacteria

The 3-CBA degrading strains were isolated using an enrichment culture technique as previously described [16]. A total of ten soil samples were collected from nearby areas on the Shizuoka University campus (34°57′30.59″ N, 138°25′32.99″ E) at Shizuoka in Japan on 23 April 2019, at 18 °C. The soil samples were stored at 4 °C in the dark until use. One gram of soil was suspended in 100 mL of BSM supplemented with 5 mM 3-CBA in 300-mL Erlenmeyer flasks and incubated at 30 °C with shaking (120 rpm) for 1 week. The culture was subsequently transferred to a newly prepared medium in order to enhance the population and this procedure was repeated three times. Afterwards, the culture was streaked on BSM supplemented with 3-CBA (0.1%, *w*/*v*) and 1.7% agar plates, followed by repeated streaking to obtain pure cultures and prevent contamination.

### 2.3. DNA Isolation and Identification Based on 16S rRNA Gene Sequencing

The bacterial isolates were identified using 16S rRNA gene sequence analysis. The total DNA of the isolated bacteria grown overnight in 3 mL of LB was extracted using a DNA isolation kit (Gentra Puregene System, Minneapolis, MN, USA) according to the manufacturer’s instructions. The amplification of 16S rRNA genes was carried out by polymerase chain reaction (PCR) with universal primer sets, 27f (5′-AGAGTTTGATCMTGGCTCAC-3′) and 1492r (5′-TACGGYTACCTTGTTACGACTT-3) [37]. The PCR was performed in a 50 µL using Go-Taq green master mix buffer (Promega, Madison, USA) using Bio-Rad MyCycler. The partial 16S rRNA gene sequences were amplified using 50 ng of total DNA and 10 pmol of each primer. The amplification of the PCR reaction is as follows: the initial step at 94 °C for 5 min; then, denaturation at 94 °C for 1 min; annealing at 58 °C for 1 min; and extension at 72 °C for 3 min for the amplification of the partial sequence of 16S rRNA genes up to 30 cycles; and finally, incubation at 72 °C for 10 min. The amplicons were purified using a QIAEX II gel extraction kit (Qiagen, Hilden, Germany) and cloned with T-vector pGEM-T-easy and *Escherichia coli* DH5α for DNA sequencing. The isolated strains were then identified by their 16S rRNA sequences using the EzBioCloud web server (https://www.ezbiocloud.net/, accessed on 30 March 2023). The 16S rRNA gene sequences were aligned using the ClustalW option implemented in MEGA v11.0 [38] and a neighbor-joining phylogenetic tree of the 16S rRNA gene sequences of the 3-CBA degraders was constructed with representative bacteria using a bootstrap test based on 1000 replicates.

### 2.4. Measurement of the Growth Efficiency of the Strains

Pre-culture of the isolates and the reference strain *Caballeronia* sp. NK8 [22] was prepared by culturing cells in 100 mL Erlenmeyer flasks containing 20 mL BSM with 3-CBA (5 mM), PC (5 mM), 3-HB (5 mM), 4-HB (5 mM) or GS (5 mM) as the sole carbon and energy source at 30 °C on a rotary shaker at 180 rpm until the mid-log phase. These pre-cultures were inoculated into 5 mL of BSM containing 3-CBA (5 mM), PC (5 mM), 3-HB (5 mM), 4-HB (5 mM) or GS (5 mM) in L-form test tubes. Bacterial growth was automatically measured by measuring the turbidity at 600 nm until the stationary phase was reached using a TVS062CA (ADVANTEC MFS, Inc., Dublin, CA, USA) compact rocking incubator with shaking at 70 rpm. BSM supplemented with 3-CBA (5 mM), PC (5 mM), 3-HB (5 mM) or GS (5 mM) without any bacterial culture were used as control. The experiments were conducted separately in triplicate and the average values were recorded in the figures and tables. Error bars in the figures represent the standard deviation of the three measurements.

The exponential maximum specific growth rate of the isolates on 3-CBA was investigated using the Monod equation (Equation (1)) [39]:*µ* = (*µ_ma_* . S)/(K_s_ + S)(1)
where *µ* defines the specific growth rate (h^−1^), *µ_max_* is the maximum specific growth rate (h^−1^), S is the growth substrate concentration at time *t* and K_s_ denotes the half-saturation coefficient.

The specific growth rate (µ, h^−1^) of the isolates was calculated using Equation (2) by plotting ln (dX) vs. time [40]:*µ*X = dX/dT(2)
where X is the OD_600_ value of the isolates at time t (h) and *µ* is calculated by integrating Equation (2) with conditions, X = X_0_ at *t* = *t*_0_ [41].

Doubling time (td) was determined using Equation (3):td = (ln2)/*µ*.(3)

### 2.5. Estimation of 3-Chlorobenzoate Degradation by the Isolates and Metabolite Assay

The concentration of 3-CBA was measured using high-performance liquid chromatography (HPLC) according to the method described previously [42]. For the HPLC analysis, 300 μL of isolated cell culture strains were collected and 100 μL methanol was added to stop bacterial growth. The mixture was vortexed and then centrifuged at 9100× *g* for 3 min at 4 °C. The supernatant was filtered through a 0.2 μm pore-size hydrophilic polytetrafluoroethylene (PTFE) membrane filter (Merck KGaA, Darmstadt, Germany) and then subjected to HPLC analysis (Jasco, Tokyo, Japan) in a Unifinepak C18 column (2.0 mm ID × 150 mmL, 3 μm; Jasco, Japan). Water-acetonitrile-acetic acid was used as the mobile phase to analyze 3-CBA (45:50:5, *v*/*v*). The flow rate was 0.2 mL min^−1^ and the column temperature was maintained at 25 °C. A UV-2075 wavelength detector (Jasco) was used to detect 3-CBA at 190 nm. Chlorobenzoate concentrations were calculated using a standard curve. The degradation experiments were conducted for 28 h and samples were collected after 8, 16, 20, 24 and 28 h to measure the 3-CBA concentration. The experiments were performed independently in triplicate to ensure the reproducibility of the findings. The 3-CBA degradation rate was calculated by dividing the change in 3-CBA concentration by the change in time over a specific interval.

The resultant two metabolite peak solvents from each isolate were collected from the HPLC and identified using a mass spectrometry (MS) instrument (microOTOF, Bruker, Billerica, MA, USA) equipped with an electrospray ionization source and operated in negative polarity mode.

### 2.6. Genomic DNA Extraction, Sequencing, Assembly and Annotation

The genomic DNA of the isolated strains was extracted using NucleoSpin Tissue (Macherey-Nagel, Duren, Germany). The draft genome sequences of the six isolates were determined using the HiSeq 2500 platform (Illumina, San Diego, CA, USA). The paired-end raw reads (2 × 151 bp) were cleaned using Trimmomatic v.0.39 [43]. Adapter sequences, the terminal 151 bases, low-quality reads of <Q15 and reads of less than 140 bp were trimmed. The resulting high-quality reads were assembled using SPAdes genome assembler version 3.15.2 [44]. The annotation of genes in each genome was performed using DFAST v.1.2.11 (https://dfast.nig.ac.jp, accessed on 2 June 2021) [45] using an in-house database containing genome sequences of *Burkholderia*, *Caballeronia*, *Cupriavidus* and *Paraburkholderia* strains and plasmids in the NCBI RefSeq database on 2 June 2021.

### 2.7. Bioinformatics

BlastKOALA [46] was used for the functional characterization of the draft genome to identify the presence of 3-CBA degradative genes using the Kyoto Encyclopedia of Genes and Genomes (KEGG) database [47]. It assigned a K number to each coding sequence (CDS), which is the number in the database of molecular functions represented in terms of functional orthologs. Easyfig version 2.2.2 [48] was then used to compare the genomic sequences of the isolates and the reference strain. The average nucleotide identity (ANI), average amino acid identity (AAI) and percentage of conserved protein (POCP) were calculated to assess the genetic similarity between the obtained isolates and the closest strains using the Ortho ANI Calculator in EzBioCloud (https://www.ezbiocloud.net/tools/ani, accessed on 30 march 2023) [49], average AAI using the EzAAI tool (http://leb.snu.ac.kr/ezaai, accessed on 20 April 2023) [50] and POCP as described by Qin et al. [51].

### 2.8. Deposition of Nucleotide Sequences

The draft genome sequences of the six isolates were deposited in the DNA Database of Japan (DDBJ) and were assigned accession numbers for each isolate. The accession numbers for each isolate’s contigs are as follows:BPUS01000001-BPUS01000112 for 19CS4-2BPUQ01000001-BPUQ01000086 for 19CS1-1BPUR01000001-BPUR01000082 for 19CS2-2BPUT01000001-BPUT01000118 for 19CS9-1BPUP01000001-BPUP01000125 for 19C8BPUO01000001-BPUO01000038 for 19C6.

## 3. Results and Discussion

### 3.1. Isolation and Identification of 3-Chlorobenzoate Degrading Bacteria

Twenty-two bacterial colonies were obtained from enriched cultures using carbon-free BSM supplemented with 5 mM 3-CBA as the sole carbon source. Six strains were successfully isolated from the enriched cultures. The isolates designated as strains 19CS4-2, 19CS2-2, 19CS1-1, 19CS9-1, 19C8 and 19C6 were identified as members of genera *Caballeronia*, *Paraburkholderia* or *Cupriavidus* (Table 2), based on their partial 16S rRNA gene sequence identities (97.62–99.17%) (Appendix A).

### 3.2. Measurement of Growth Efficiency with 3-CBA of the Isolated Strains

The growth of the isolates (19CS4-2, 19CS9-1, 19CS1-1, 19CS2-2, 19C8 and 19C6) was assessed in the presence of 3-CBA as the carbon source by monitoring turbidity at 600 nm until they reached the stationary phase. Notably, three isolates, 19CS4-2, 19CS9-1 and 19CS1-1, displayed significantly faster growth compared with that of the remaining three isolates and reference strain NK8 (Figure 1). The data are represented as the mean ± standard deviation of three independent experiments.

The growth kinetics were calculated using the Monod kinetic model by plotting the graph ln(OD_600_) vs. incubation time (Appendix A). In Appendix A, the slope of the curve shows the maximum specific growth rate of the isolates (Equation (2)). The exponential growth rate and doubling time comparison among the isolates and the reference strain *Caballeronia* sp. NK8 are shown in Table 3. The maximum specific growth rates and doubling times varied among the isolates and the reference strain. The strains 19CS4-2 and 19CS1-1 exhibited the highest maximum specific growth rate (*µ_max_*) of 0.30 h^−1^ and 0.29 h^−1^, respectively, along with the shortest doubling time when compared to the remaining four isolates and the reference strain NK8. The maximum specific growth rate of 19CS9-1 was very close to that of the NK8 strain. These six isolates were then classified into three groups based on their genera. The first group consisted of *Caballeronia* genera (19CS4-2, 19CS2-2 and 19CS1-1); the second group included *Paraburkholderia* genera (19CS9-1 and 19C8); and the third group included *Cupriavidus* genera (19C6). One representative strain from each group (genus *Caballeronia* strain 19CS4-2, genus *Paraburkholderia* 19CS9-1 and *Cupriavidus* 19C6) was selected for further evaluation of their degradation ability and genetic analysis based on their faster growth than the other strains.

### 3.3. Biodegradation of 3-Chlorobenzoate by Three Bacterial Strains Isolated from the Enrichment Culture

HPLC analysis was performed on the three representative strains based on their growth curves at each time point. The samples were collected at 8, 16, 20, 24 and 28 h after the start of the incubation period. In Appendix A, two HPLC chromatogram results (after 8 h and a complete reduction of 3-CBA) are shown to illustrate the changes in 3-CBA concentration over time. The analysis showed that 19CS4-2 and 19CS9-1 reduced 5 mM 3-CBA levels to below the detection limit within 20 h of cultivation, whereas 19C6 required 28 h to achieve the same level of degradation (Figure 2). The data is represented as the mean ± standard deviation of three independent experiments. The degradation rate of the isolates 19CS42, 19CS9-1 and 19C6 were 0.29 mM h^−1^, 0.23 mM h^−1^ and 0.10 mM h^−1^, respectively. 19CS4-2 had a higher degradation rate than the other isolates. The 19C6 isolate was able to degrade 3-CBA with a low growth rate. The low growth rate in 19C6 might be attributable to the presence of a toxic intermediate during the degradation of 3-CBA. The accumulation of these toxic compounds can hinder bacterial growth and metabolic activity [21]. The metabolites were then investigated by MS analysis.

HPLC analysis revealed two metabolites in all isolates grown on 3-CBA (Figure 3). The two peaks of the isolates were collected 40 h after the start of the incubation period. The mass spectra of the isolates showed deprotonated molecular ions at *m*/*z* = 157.02 [M−H]^−^ and *m*/*z* = 174.99 [M−H]^−^. These ions correspond to the molecular ions [M^−^] at *m*/*z* = 158.02 and *m*/*z* = 175.99, which are the masses of maleylacetate and chloro-*cis*,*cis*-muconate, respectively (Figure 3). The metabolites were found in all isolates, suggesting they might be involved in the 3-CBA degradation pathway. Although the metabolites were found in all strains, the growth characteristics of strain 19C6 were notably different to those of 19CS4-2 and 19CS9-1. Further genomic analyses were conducted to gain a more detailed understanding of the genetic mechanisms underlying the different degradation abilities of these isolates.

### 3.4. Genomic Analyses

#### 3.4.1. Draft Genome Sequences of Three Isolates

The draft genome sequences of the three isolated strains were used to predict the pathway(s) by which they metabolized 3-CBA. The draft sequences of strains 19CS4-2, 19CS9-1 and 19C6 were 9.05, 9.42 and 6.80 Mb, respectively, consisting of 112, 118 and 38 contigs, respectively; and their G + C content was 62.5%, 62.5% and 64.9%, respectively (Table 4). From the comparative analysis of annotation of whole genome sequences of each strain, two isolates, 19CS4-2 and 19CS9-1, were predicted to contain plasmids, whereas isolate 19C6 was not. The presence or absence of plasmids in these isolates may have significant implications for their genetic diversity, adaptability and potential interactions within their respective ecological niches.

The isolates were identified based on whole-genome sequencing data providing a comprehensive taxonomic classification approach. The AAI serves as a valuable metric for taxonomic classification and provides a quantitative measure for determining the separation of prokaryotic genera [50,52,53]. Another effective method for genus delineation of prokaryotes is the POCP [51]. In addition, ANI measures the average percentage of nucleotide sequence similarity between two genomes and is commonly used for species delineation [49,54,55]. The 19CS4-2, 19CS9-1 and 19C6 strains had AAI values of 93.3%, 95.0% and 86.9%, respectively; POCP values of 78.6%, 82.0% and 75.5%, respectively; and ANI values of 91.7%, 92.7% and 83.6%, respectively, identity with the closet strains, including *Caballeronia novacaledonica* LMG 28615(T), *Paraburkholderia hospita* LMG 20598(T) and *Cupriavidus pauculus* CCUG 12507(T), respectively (Table 4 and Appendix A). These results also suggest that 19CS4-2, 19CS9-1 and 19C6 could be assigned to the genera *Caballeronia*, *Paraburkholderia* and *Cupriavidus*, respectively.

#### 3.4.2. Determination of the Genes Involved in the 3-CBA Degradation Pathways

Based on the draft genome sequences of the three isolates, their degradation pathways were predicted using BlastKOALA, an automatic annotation server accessible on KEGG web site (http://www.kegg.jp/blastkoala/). All the putative genes for degradation were located in chromosomal contigs. Putative genes for 3-CBA degradation via the CC degradation pathway were found in the genome of the isolates (see below and Appendix A).

Regarding the CC pathway, the degradation of 3-CBA can be initiated by dioxygenase enzymes, which include a Rieske-type (2Fe-2S) terminal oxygenase component and nicotinamide adenine dinucleotide (NADH): acceptor reductase component and convert 3-CBA into a dihydrodiol derivative and then, convert the intermediate compound into CC with the release of CO_2_. The three isolates possessed *cbeABCD* genes, which are probably involved in the conversion of 3-CBA to CC by the action of (chloro)benzoate dioxygenase and dehydrogenase enzymes (the CC pathway) (Figure 4a). The *cbe* genes of 19CS4-2, 19CS9-1 and 19C6 showed 89–94%, 60–75% and 77–86% identity, respectively, with the reference strain *Caballeronia* sp. NK8 (AP024325; located in the plasmid pNK81) (Figure 5a). However, only two strains, 19CS4-2 and 19CS9-1, possessed *tfdCDEF* gene clusters, which were probably involved in the conversion of CC to 3-oxoadipate (Figure 4a). The *tfd* gene is important for the degradation of chloroaromatic compounds and wild-type 3-CBA-degrading bacteria have a limited number of the *tfd* genes [23,56]. The *tfd* genes in strains 19CS4-2 and 19CS9-1 showed 98–100% and 81–86% identity with those in the reference genome NK8 (AP024328; located in the plasmid pNK84) (Figure 5a). 19CS4-2 and 19CS9-1 showed varying degrees of identity with NK8. The presence of *cbe* and *tfd* genes in 19CS4-2 and 19CS9-1 indicates that they may metabolize 3-CBA via the CC pathway. Notably, 19C6, which showed a slower degradation rate than strains 19CS4-2 and 19CS9-1, did not possess the *tfdCD* genes involved in the conversion of CC to dienelactone. The *tfdEF* gene 19C6 showed low identity (32–53%) with that of the reference genome NK8 (Figure 4a and Figure 5a). Two degradation products, maleylacetate and chloro-*cis*,*cis*-muconate, which are the intermediates of the CC *ortho*-cleavage pathway, were found in all isolates. The slow growth of 19C6 might have been due to the lack of the *tfdCD* genes involved in the degradation of CC and chloro-*cis*,*cis*-muconate, which are intermediates in the degradation pathway of 3-CBA. These products might inhibit the growth of the bacteria [21]. Consequently, even at low-optical-density (OD) values, strain 19C6 is capable of reducing the concentration of 3-CBA, possibly owing to the presence of *cbe* genes responsible for the initial steps of 3-CBA degradation. This suggests that the CC degradation mechanism of 19C6 could differ from those of the other two strains.

All three isolates possessed putative *catA* genes encoding catechol dioxygenase, which may be involved not only in catechol metabolism, but also in CC degradation [17,57]. The *catA* genes of 19CS4-2 and 19C6 were located upstream of the *cbeABCD* gene cluster and probably formed an operon (Figure 5a). In contrast, the *catA* gene of the 19CS9-1 strain was at a separate location from the *cbeABCD* gene clusters (Figure 5a), suggesting that the expression of the *catA* gene of 19CS9-1 might be regulated differently from those of the other two strains.

In contrast, some 3-CBA degraders (e.g., *Alcaligenes* sp. L6 and *Bacillus* sp. OS13) have been predicted to metabolize 3-CBA via PC and/or GS rather than via CC [30,31,32,34]. In these pathways, the intermediate compounds include 4-hydroxybenzoate (4-HB), 3-hydroxybenzoate (3-HB), protocatechuate (PC) and gentisate (GS). PC and GS are thought to be converted from 4-HB or 3-HB [30,31,32,34]. However, the initial enzyme that converts 3-CBA to 4-HB or 3-HB has not yet been identified. Putative genes involved in the metabolism of 4-HB and 3-HB via PC and/or GS were found in the genomes of the three experimental isolates (see below and Appendix A).

In most bacteria, 4-HB is metabolized by 4-HB-3-monooxygenase, resulting in the formation of PC (the PC pathway) (Figure 4b), which serves as a central intermediate [58,59]. 4-HB-3-monooxygenase is encoded by the *pobA* gene in *Pseudomonas* strains [60,61]. Genomic analysis revealed that all three isolates possessed the *pobA* gene. The *pobA* genes of 19CS4-2, 19CS9-1 and 19C6 showed 94%, 65% and 60% identity, respectively, with those of NK8 (AP024323; located on chromosome 2) (Figure 5b). A variety of diverse *pca* genes with a capacity for PC degradation have been found in bacteria [58,59,62]. The three experimental isolates also possessed *pca* genes, which are probably involved in the conversion of PC to succinyl-CoA and acetyl-CoA (Figure 4b). In 19CS4-2, the *pobA* gene was located separately from the *pca* genes, whereas the genes were located together in 19CS9-1 and 19C6, probably forming an operon. Furthermore, the *pcaG* and *pcaH* genes, encoding the putative alpha- and beta-subunits of PC-3,4-dioxygenase for PC degradation, were located in regions distant from other *pca* gene clusters in 19CS4-2 and 19CS9-1 (Figure 5b). In contrast, *pcaG* and *pcaH* were located with the other *pca* genes in 19C6, probably forming an operon (Figure 5b). The *pca* genes in 19CS4-2, 19CS9-1 and 19C6 showed 92–96%, 65–88% and 62–76% identity, respectively, with those in NK8 (AP024322; and AP024323, located on chromosome 1 and 2, respectively) (Figure 5b). Therefore, all strains may have a PC route for 4-HB degradation, although the genetic structures of the *pob* and *pca* genes are different from one another (Figure 5b). These findings suggest that 19CS4-2, 19CS9-1 and 19C6 possess putative genes involved in 4-HB metabolism via PC.

Genetic analyses of the GS pathway revealed that 19CS4-2 and 19CS9-1 did not possess any putative genes encoding enzymes for 3-HB degradation. In contrast, 19C6 possessed an *mhbM* gene, which probably encodes 3-HB-6-monooxygenase, an enzyme that converts 3-HB to GS (GS pathway) (Figure 4b). Moreover, 19C6 also possessed *mhbD*, *mhbH* and *mhbI* genes, which probably encode GS-1,2-dioxygenase, maleylpyruvate isomerase and fumarylpyruvate hydratase, respectively (Figure 4b). These enzymes may be involved in converting gentisate to fumarate or pyruvate. The *mhb* genes in 19C6 showed 80–86% identity with those of another reference strain, *Cupriavidus necator* NH9 (CP017758; located on chromosome 2) (Figure 5c). These findings suggest that 19C6 possesses putative genes involved in 3-HB metabolism via GS. The identification of the GS catabolic pathway is known to be present only in a limited number of members of the *Burkholderiaceae* family [58]. The identification of the GS catabolic pathway in isolate 19C6 highlights its unique metabolic capabilities and distinguishes it from the other two experimental isolates (19CS4-2 and 19CS9-1).

### 3.5. Growth of the Isolates on Protocatechuate, Gentisate, 4-Hydroxybenzoate and 3-Hydroxybenzoate

Genomic analysis of the three experimental isolates showed that they contained 4-HB and 3-HB degradative genes via PC or GS as intermediate compounds. First, growth assays were performed using either 4-HB or 3-HB as the sole carbon source. All three isolates grew on 4-HB or 3-HB (Appendix A). Next, growth assays were performed using PC (5 mM) or GS (5 mM) as a carbon source. All three strains grew on PC (Appendix A), whereas only 19C6 grew on GS (Appendix A). The growth studies indicated that 4-HB degradation via PC is present in all three strains. The studies also indicated that a 3-HB degradation via GS is present in 19C6, which has also been observed in other bacteria, including *K. pneumoniae* M5a1 [63], *Corynebacterium glutamicum* RES167 [64] and *Rhodococcus* sp. strain NCIMB 12038, in which GS has been described as the main ring-cleavage pathway for 3-HB assimilation [65]. It is possible that 19CS4-2 and 19CS9-1 converted 3-HB into other intermediate compounds. The findings from the reference strain *C. necator* NH9 provide additional insights that can be related to our studies. *C. necator* NH9 grows on 3-HB and 4-HB [42,66] and shows transcriptional induction of putative genes involved in 3-HB degradation when grown on 3-CBA as the sole carbon source [42,66]. This suggests that NH9 may have a GS pathway. In contrast, the reference strain *Caballeronia* sp. NK8 did not grow on 3-HB (Appendix A). Although the experimental isolates, including 19CS4-2 and 19CS9-1, lacked specific 3-HB degradation genes, they were still able to grow on 3-HB. This suggests that these strains might employ alternative or unidentified pathways for 3-HB conversion. The similarities of the results of our studies to those of studies using *C. necator* NH9 and *Caballeronia* sp. NK8 underscore the importance of investigating the metabolic versatility and diversity within bacterial species. The results also show that different strains within the same bacterial genus may exhibit distinct metabolic capabilities and employ different metabolic pathways for the degradation of specific compounds.

The genome-wide comparisons of the experimental isolates with the reference strains revealed significant differences between them. The observed differences in the genomes of the isolates compared with those of the reference strains open up new possibilities for exploration. By studying these differences, it is possible to gain a deeper understanding of the metabolic versatility of the isolates and these findings highlight the importance of further research to fully understand the degradation pathways of these organisms.

## 4. Conclusions

In this study, we isolated three 3-CBA degraders with different growth and degradation rates from the soil. The three isolates belonged to the genera *Caballeronia*, *Paraburkholderia* and *Cupriavidus*. Genetic analyses revealed their genetic diversity and functional differences involved in 3-CBA degradation. These findings expand our knowledge regarding the involvement of multiple degraders in the biodegradation process of 3-CBA within the environment. These degraders can adapt to different environmental conditions. These findings highlight the potential ecological significance of isolates obtained from the natural environment. Future research could focus on elucidating the specific mechanisms and regulatory networks involved in the degradation pathways.

## Figures and Tables

**Figure 1 microorganisms-11-01684-f001:**
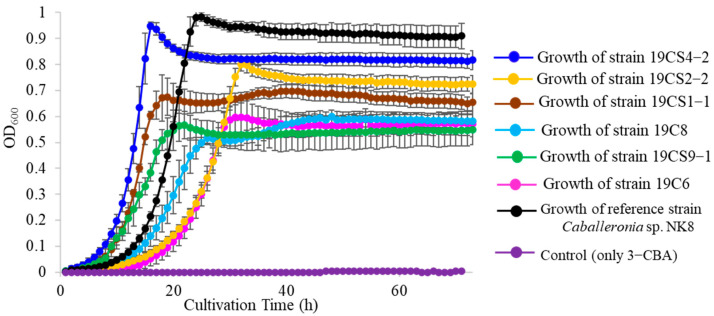
The cell growth of isolates 19CS4-2, 19CS2-2, 19CS1-1, 19CS9-1, 19C8 and 19C6 with reference *Caballeronia* sp. NK8, in basal salt medium (BSM) supplemented with 5 mM 3-chlorobenzoate (3-CBA). The data are represented as the mean ± standard deviation of three independent experiments. Error bars that are not visible lie within the data point.

**Figure 2 microorganisms-11-01684-f002:**
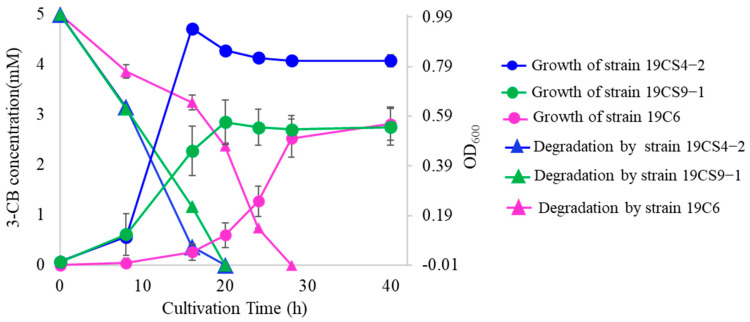
The growth and 3-chlorobenzoate (3-CBA) degradation rates of strains 19CS4-2, 19CS9-1 and 19C6 in BSM supplemented with 5 mM 3-CBA. The data is represented as a mean ± standard deviation of three independent experiments. Error bars that are not visible lie within the data point.

**Figure 3 microorganisms-11-01684-f003:**
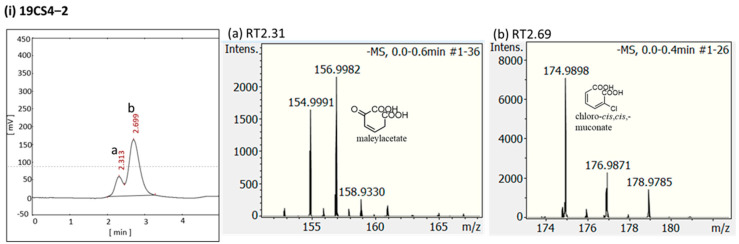
High-performance liquid chromatography (HPLC) and mass spectroscopy (MS) identification of the metabolites during degradation of 3-chlorobenzoate (3-CBA) by (**i**) 19CS4-2, (**ii**) 19CS9-1 and (**iii**) 19C6. The mass spectra show major deprotonated molecular ions at *m*/*z* = 157.002 [M−H]^−^ (**a**) and *m*/*z* = 174.99 [M−H]^−^ (**b**). These compounds were identified as maleylacetate and chloro-*cis*,*cis*-muconate, respectively, in all the isolates.

**Figure 4 microorganisms-11-01684-f004:**
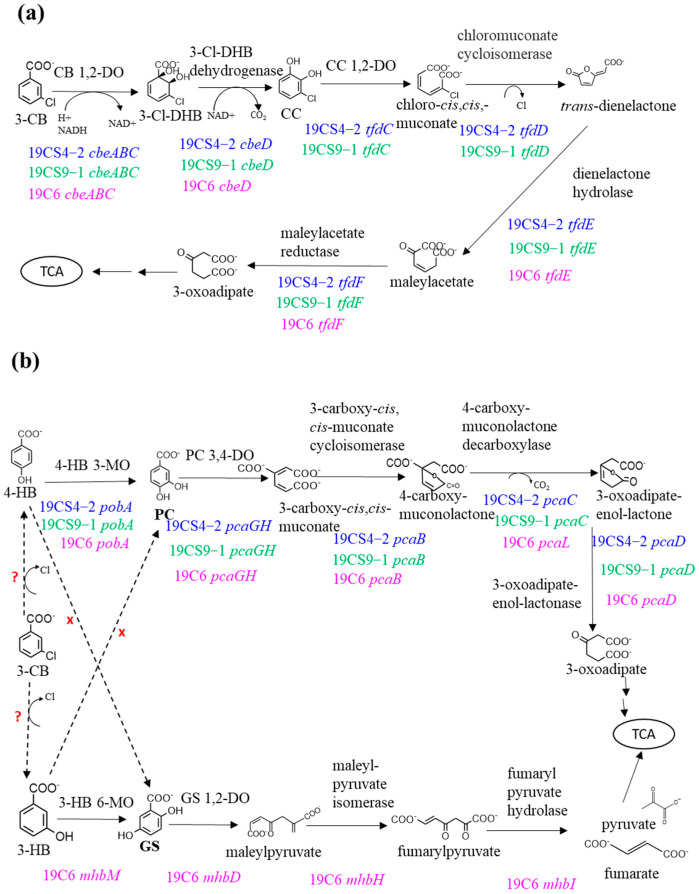
Putative 3-chlorobenzoate (3-CB) degradation via chlorocatechol (CC) *ortho*-cleavage pathways involved in the isolates (**a**). Putative 4-HB and 3-HB degradation via PC and GS in the isolates, respectively (**b**). The strains 19CS4-2,19CS9-1 and 19C6 are colored in blue, green and pink, respectively. The enzyme is unknown in the conversion of 3-CBA to 4 or 3-HB. Abbreviations: 3-CB, 3-chlorobenzoate; 3-Cl-DHB, chloro-3,5-cyclohexadiene-1,2-diol-1-carboxylate; CC, chlorocatechol; DO, dioxygenase; GS, gentisate; HB, hydroxybenzoate; MO, monooxygenase, PC, protocatechuate.

**Figure 5 microorganisms-11-01684-f005:**
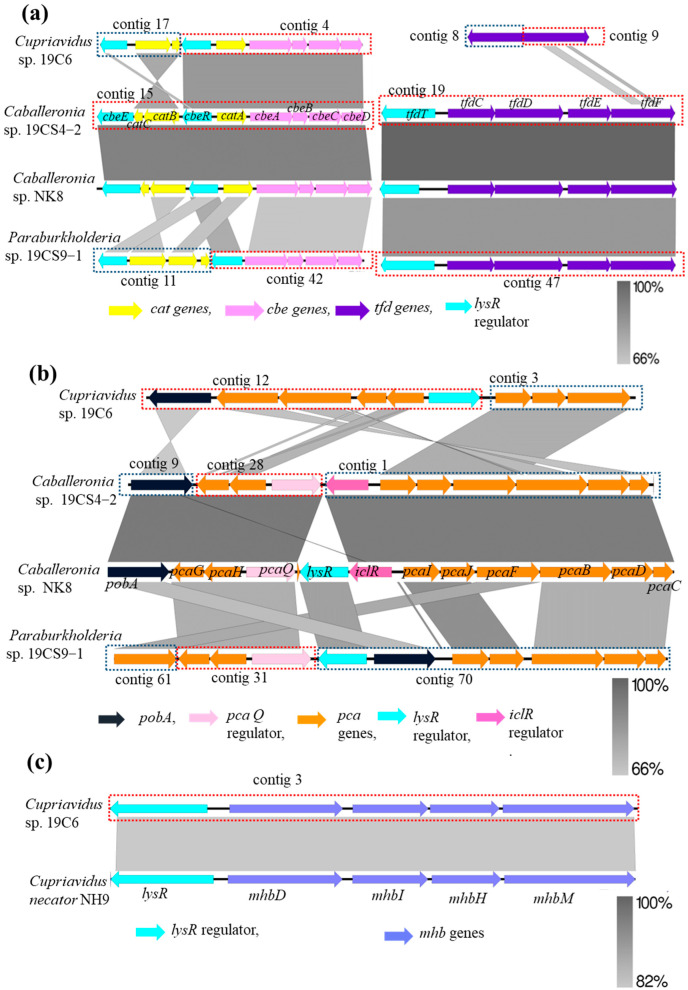
Comparison of strains 19CS4-2, 19CS9-1 and 19C6 involved in 3-CBA degradation via chlorocatechol (CC) (**a**), compared with corresponding clusters of *Caballeronia* sp. NK8 genes. Comparison of strains 19CS4-2, 19CS9-1 and 19C6 involved in 4-hydroxybenzoate degradation via protocatechuate (PC) compared with corresponding clusters of *Caballeronia* sp. NK8 genes (**b**). Comparison of clusters of strains 19C6 involved in 3-hydroxybenzoate degradation via gentisate (GS) (**c**), compared with corresponding clusters of *Cupriavidus necator* NH9 genes. Red and blue borders represent the contig number, where clusters are present in the bacterial genome.

**Table 2 microorganisms-11-01684-t002:** Results of the EzBioCloud homology search using the partial 16S rRNA gene sequences of the strains obtained in the experiment.

Strain	Top Hit Taxon	Identity (%)	Completeness (%)
19CS4-2	*Caballeronia novacaledonica* LMG 28615(T)	97.62	100
19CS2-2	*Caballeronia novacaledonica* LMG 28615(T)	97.62	100
19CS1-1	*Caballeronia novacaledonica* LMG 28615(T)	97.62	100
19CS9-1	*Paraburkholderia hospita* LMG 20598(T)	98.62	100
19C8	*Paraburkholderia hospita* LMG 20598(T)	99.17	100
19C6	*Cupriavidus pauculus* LMG 3413(T)	98.86	100

**Table 3 microorganisms-11-01684-t003:** Comparison of the exponential maximum specific growth rate and doubling time among the isolates and the reference strain *Caballeronia* sp. NK8.

Features	19CS4-2	19CS1-1	19CS2-2	19CS9-1	19C8	19C6	NK8
Maximum specific growth rate (*µ_max_*) [h^−1^]	0.30	0.29	0.16	0.25	0.18	0.18	0.24
Doubling time (td) [h]	2.30	2.35	4.32	2.66	3.69	3.70	2.77

**Table 4 microorganisms-11-01684-t004:** Summary of genome sequences of the isolates.

Isolates	Putative Genus	N_50_ bp	#Contig	Size (Mb)	GC (%)	#CDS	Mean-Coverage ^1^	AAI (%)	POCP (%)	Ortho- ANIu	Plasmid
19CS4-2	*Caballeronia*	274,011	112	9.05	62.5	8241	101.0	93.3	78.6	91.7	Yes
19CS9-1	*Paraburkholderia*	257,300	118	9.41	62.5	8371	66.7	95.0	82.0	92.7	Yes
19C6	*Cupriavidus*	483,946	38	6.80	64.9	6067	127.9	86.9	75.5	83.6	No

^1^ Total size of filtered reads (bp)/genome size (bp). Abbreviations: AAI, average amino acid identity; ANI, average nucleotide identity; CDS, coding sequence; GC, guanine–cytosine; OrthoANIu, OrthoANIu tool, a standalone average nucleotide identity (ANI) calculator; POCP, percentage of conserved protein.

## Data Availability

Accession number of each isolate: BPUS01000001-BPUS01000112 for 19CS4-2, BPUQ01000001-BPUQ01000086 for 19CS1-1, BPUR01000001-BPUR01000082 for 19CS2-2, BPUT01000001-BPUT01000118 for 19CS9-1, BPUP01000001-BPUP01000125 for 19C8, BPUO01000001-BPUO01000038 for 19C6.

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
