# Peer review of "Isolation and Genomic Analysis of 3-Chlorobenzoate-Degrading Bacteria from Soil"

_microorganisms, 2023, doi:10.3390/microorganisms11071684_

Round 1
Reviewer 1 Report
microorganisms-2413234
Authors
Comments
The research is well organized and structured
I have no content comments
Author Response
Responses to the comments of Reviewer 1:
The research is well organized and structured
I have no content comments
RESPONSE: Thank you for your positive feedback on the organization and structure of our research. We greatly appreciate your assessment.
Reviewer 2 Report
Ara et al. report on the isolation and genome analysis of several new 3-chlorobenzoate-degrading bacterial strains. Unfortunately, the study is only based on basic growth experiments and speculations on the possible pathways based on the genome sequences, but does not provide any real data to substantiate the hypothetic pathways. The manuscript does not contain any novel data besides the genome sequences, which unfortunately does not justify its publication in the current state.
Lines:
23 investigates
64 mind the grammar: “these compounds … and therefor they are known as … pathways” needs a new subject. These errors occur too often in the manuscript to note them all
68 if the authors were interested in new degradtion pathways (by “different degraders”), why bother with members of the same genera using the same pathways as previously known?
80ff the chapters on bacterial isolation, DNA isolation and sequencing contain lengthy, but rather trivial methods explanations, which can be significantly condensed.
120 what is meant with “culturing BSM”???
175 16S rRNA analysis provides a probable affiliation of these strains, but they cannot be formally identified only based on this technique. To actually determine the taxonomic status, DNA-DNA hybridization (or alternatively a full-genome comparison with reference strains) is required
180 the different growth phenotypes in Fig. 1 may be due to different lag times. Otherwise, the data seem to be hardly interpretable, because the data points often do not cover the exponential phase (see the dark blue, brown, and green curves). Data within the exp- phase would be necessary for calculating growth rates. Moreover, the dark blue curve (19CS4-2) jumps directly to max. OD from almost zero, then the OD declines. The authors also do not indicate how reproducible the data were, they rather seem to re-use the exact same curves over and over (Fig. 2 shows the exact same curves as Fig. 1, which is impossible if a separate experiment has been performed). As it is, these experiments do not allow to draw any quantitative conclusions.
198 speculations on “different machanisms” are impossible with the data provided. The authors did not even look for potential degradation intermediates to confirm the imferred pathways.
211 what suggests the presence of plasmids?
214 what are the closest strains?
215 these abbreviations should not be used in the text. Rather, the authors should indicate what they mean (e.g. are they providing anything on the overall genome relatedness with reference strains?)
225 genes for degradation? What pathways, which enzymes?
226 the pathway is hypothetical as long as there are no other indications for its presence
231 Rieske .. and NADH are no enzymes!
232 the enzymes should be named
234ff is the identity given based on genes or gene products ? (DNA identities have no bearing for the function of enzymes)
252 different location is no proof for different regulation
All the rest of the results consists of wild speculation or lengthy description of pathways in other known strains where potential genes are either present (pobA) or not (PC or GS pathways). Since no experiments have been done (biochemical, proteomic, RNASeq, …) none of the speculations is worth publishing at this state.
Problems include the differentiation of genes and gene products, or proper relation of entities (e.g. Rieske and NADH are not proteins, where you talk of pathways or of enzymes...). Moreover, the use of tenses need to be improved (e.g. use present instead of past tense when describing genome content, pathway function, etc.).
Author Response
Responses to the comments of Reviewer 2:
Ara et al. report on the isolation and genome analysis of several new 3-chlorobenzoate-degrading bacterial strains. Unfortunately, the study is only based on basic growth experiments and speculations on the possible pathways based on the genome sequences, but does not provide any real data to substantiate the hypothetic pathways. The manuscript does not contain any novel data besides the genome sequences, which unfortunately does not justify its publication in the current state.
RESPONSE: Thank you for reviewing our work. We have rewritten the manuscript to describe the new findings in the present study clearly. We have also added experiments to identify intermediates and provide evidence of the degradation pathway in our isolated bacterial strains.
23 investigates
RESPONSE: Corrected. (line 23 in the revised manuscript)
64 mind the grammar: “these compounds … and therefore they are known as … pathways” needs a new subject. These errors occur too often in the manuscript to note them all
RESPONSE: Corrected. (line 78 in the revised manuscript)
68 if the authors were interested in new degradation pathways (by “different degraders”), why bother with members of the same genera using the same pathways as previously known?
RESPONSE: Thank you very much for the comment. At the beginning, we did not necessarily have an interest in the new degradative pathways. Based on the comparisons of the growth rates of the isolates, we found differences in them. Then, we performed in-depth analyses of the isolates using genomic sequences.
80ff the chapters on bacterial isolation, DNA isolation and sequencing contain lengthy, but rather trivial methods explanations, which can be significantly condensed.
RESPONSE: According to the reviewer’s comment, we have rewritten the method section for bacterial isolation (lines 99-104 in the revised manuscript)
120 what is meant with “culturing BSM”???
RESPONSE: We have rewritten this part as follows: “Basal salt medium (BSM) supplemented with 3-CB (5 mM), PC (5 mM), 3-HB (5 mM), or GS (5 mM) without any bacterial culture were used as control.”. (lines 135-136 in the revised manuscript)
175 16S rRNA analysis provides a probable affiliation of these strains, but they cannot be formally identified only based on this technique. To actually determine the taxonomic status, DNA-DNA hybridization (or alternatively a full-genome comparison with reference strains) is required
RESPONSE: Thank you very much for your comment. We agree with the comment that the taxonomic identification of bacteria based on the comparisons of 16S rRNA gene sequences is not sufficient for taxonomic status. In the present study, we have already performed genome-wide analyses to determine the taxonomic status of the isolates, including average nucleotide identity (ANI), amino acid identity (AAI), and percentage of conserved protein (POCP) using the genome sequences of each isolate and the reference strains. (Table 4 and Lines 294-304 in the revised manuscript).
180 the different growth phenotypes in Fig. 1 may be due to different lag times. Otherwise, the data seem to be hardly interpretable, because the data points often do not cover the exponential phase (see the dark blue, brown, and green curves). Data within the exp- phase would be necessary for calculating growth rates. Moreover, the dark blue curve (19CS4-2) jumps directly to max. OD from almost zero, then the OD declines. The authors also do not indicate how reproducible the data were, they rather seem to re-use the exact same curves over and over (Fig. 2 shows the exact same curves as Fig. 1, which is impossible if a separate experiment has been performed). As it is, these experiments do not allow to draw any quantitative conclusions.
RESPONSE: Thank you very much for your comments. We have replaced Figure 1 with a new one with each data point. Regarding the reproducibility of these growth curves, we conducted independent triplicate experiments. Error bars are included to indicate the variability within the data sets. To avoid confusion, we have rewritten the method (lines 136-138) and result sections to clearly show that we used the independent methods in the revised manuscript. We have also added the growth rate result in Table 3 and the degradation rate in Line 256-257in the revised manuscript.
198 speculations on “different machanisms” are impossible with the data provided. The authors did not even look for potential degradation intermediates to confirm the imferred pathways.
RESPONSE: According to the reviewer’s comment, we have rewritten this part as follows: “The 19C6 isolate was able to degrade 3-CBA with a low growth rate. The low growth rate in 19C6 might be attributable to the presence of a toxic intermediate during the degradation of 3-CBA. The accumulation of these toxic compounds can hinder bacterial growth and metabolic activity” (Line 259-261). We have also added the identification of the intermediates shown in Figure 3 in the revised manuscript.
211 what suggests the presence of plasmids?
RESPONSE: Currently, the reason(s) and effect(s) of the presence of plasmids in 3-CBA degrader is not clear, although putative degradative genes were not found in these plasmids.
214 what are the closest strains?
RESPONSE: Here, we showed the results of genome-wide comparisons with the closest type strains in each genus, Caballeronia, Paraburkholderia, and Curpriavidus. To avoid confusion, we have written this part as follows: “The isolates were identified based on whole-genome sequencing data, which provide a comprehensive approach for taxonomic classification. The AAI serves as a valuable metric for taxonomic classification and provides a quantitative measure for determining the separation of prokaryotic genera [50,52,53]. Another effective method for genus delineation of prokaryotes is the POCP [51]. In addition, ANI measures the average percentage of nucleotide sequence similarity between two genomes and is commonly used for species delineation [49,54,55]. The 19CS4-2, 19CS9-1, and 19C6 strains had AAI values of 93%, 95%, and 87%, respectively; POCP values of 78%, 82%, and 75%, respectively; and ANI values of 91%, 92%, and 83%, respectively, identity with the closet strains including Caballeronia novacaledonica LMG 28615(T), Paraburkholderia hospita LMG 20598(T) and Cupriavidus pauculus CCUG 12507(T), respectively (Tables 4, S1, S2, and S3)”. (Lines 294-304 in the revised manuscript).
215 these abbreviations should not be used in the text. Rather, the authors should indicate what they mean (e.g. are they providing anything on the overall genome relatedness with reference strains?)
RESPONSE: Although ANI (average nucleotide identity), AAI (average amino acid identity), and POCP (percentage of conserved protein) are well-known values for genome-wide comparisons, we have added explanations for them and their reference in the revised manuscript (Lines 294-304 in the revised manuscript).
225 genes for degradation? What pathways, which enzymes?
RESPONSE: According to the reviewer’s comment, we have rewritten this sentences as follows: “. Putative genes for 3-CBA degradation via chlorocatechol (CC) degradation pathway was found in the genome of the isolate.” (Lines 314-315 in the revised manuscript).
226 the pathway is hypothetical as long as there are no other indications for its presence
All the rest of the results consists of wild speculation or lengthy description of pathways in other known strains where potential genes are either present (pobA) or not (PC or GS pathways). Since no experiments have been done (biochemical, proteomic, RNASeq, …) none of the speculations is worth publishing at this state.
RESPONSE: Although we did not have direct evidence of the presence of PC or GS pathways, we have shown whether the isolates could metabolize PC and/or GS in addition to 3HB and/or 4HB. The main findings of the present study are (i) the isolates showed different growth rates on 3CBA, (ii) genome-wide comparisons showed that they are different, which could explore potential avenues for further investigation. For the investigation of putative degradative pathways in detail would be future publications.
231 Rieske .. and NADH are no enzymes!
RESPONSE: According to the reviewer’s comment, we have rewritten this part as follows: “Regarding the CC pathway, the degradation of 3-CB can be initiated by dioxygenase enzymes, which include a Rieske-type [2Fe-2S] terminal oxygenase component and NADH: acceptor reductase component and convert 3-CB into dihydrodiols and then converted into CC.” (Lines 318-319 in the revised manuscript).
232 the enzymes should be named
RESPONSE: We have added the names of enzymes (Lines 323 in the revised manuscript).
234ff is the identity given based on genes or gene products? (DNA identities have no bearing for the function of enzymes)
RESPONSE: We showed the identity of nucleotide sequences of each gene, showing high identity (60-100%). In the revised manuscript, we have added the identity of amino acid sequences in Tables S4, S5.
252 different location is no proof for different regulation
RESPONSE: Here, we would like to claim that different operons could be regulated differently by different elements. Thus, we have rewritten this part as follows; “In contrast, the catA gene of the 19CS9-1 strain was at a separate location from the cbeABCD gene clusters (Figure 5a), suggesting that the expression of the catA gene of 19CS9-1 might be regulated differently from those of the other two strains.” (lines 350-351 in the revised manuscript).
Comments on the Quality of English Language
Problems include the differentiation of genes and gene products, or proper relation of entities (e.g. Rieske and NADH are not proteins, where you talk of pathways or of enzymes...). Moreover, the use of tenses need to be improved (e.g. use present instead of past tense when describing genome content, pathway function, etc.).
RESPONSE: The submitted manuscript was already edited by a professional academic English editing service, but according to the reviewer’s suggestions, the revised manuscript has been edited again.
Reviewer 3 Report
The idea of ​​the manuscript is interesting. A study is essential. Unfortunately, I missed the interpretations of the obtained results at work. The results of Fig. 2 seem somewhat controversial. More details are in the below.
Abstract:
There are no conclusions in the abstract, and the wording should be improved.
Introduction
67-68: The aim in the abstract should be the same as in the main text.
Results
182-183: ,, Among the isolates belonging to the same genus, 19CS4-2 in the genus Caballeronia and 19CS9-1 in the genus Paraburkholderia grew faster than the others (Figure 1).“
- in the figure it is clear that 19CS1-1 (brown curve) grown faster than 19CS9-1.
197: ,, These differences may be due to different metabolic mechanisms being used in the different isolates“. What are these mechanisms? If the author chose not to make a separate part of the discussion, the results must still be discussed in more depth and compared with the results of other researchers.
Figure 2. It is strange that when the OD600 is under 0,1 (almost 0), the concentration of 3-CB decreases by 1 mM (strain 19C6 – pink curve). What is the reason? It must be explained.
287-288: ,,These differences may be due to different metabolic mechanisms being used in the other isolates.“ If the author chose not to make a separate part of the discussion, the results must still be discussed in more depth and compared with the results of other researchers.
301-302: ,,These findings demonstrate the metabolic versatility of the isolates and highlight the importance of the further investigation to fully understand their degradation pathways.“ If the author chose not to make a separate part of the discussion, the results must still be discussed in more depth and compared with the results of other researchers.
References:
19 source: the year should be written in bold.
Author Response
Responses to the comments of Reviewer 3:
The idea of ​​the manuscript is interesting. A study is essential. Unfortunately, I missed the interpretations of the obtained results at work. The results of Fig. 2 seem somewhat controversial. More details are in the below.
RESPONSE: Thank you for your positive feedback on the manuscript and for bringing to our attention your concerns regarding the interpretation of the results, particularly related to Figure 2. To avoid confusion, we have rewritten the result section for Fig. 2 (see below). We hope that you find the revised version satisfactory and suitable for publication.
Abstract:
There are no conclusions in the abstract, and the wording should be improved.
RESPONSE: According to the reviewer’s suggestion, we have added a conclusion in the abstract (lines 35-37 in the revised manuscript).
Introduction
67-68: The aim in the abstract should be the same as in the main text.
RESPONSE: According to the reviewer’s suggestion, we have rewritten the aim in the abstract (lines 81-82 in the revised manuscript).
Results
182-183: ,, Among the isolates belonging to the same genus, 19CS4-2 in the genus Caballeronia and 19CS9-1 in the genus Paraburkholderia grew faster than the others (Figure 1).“
- in the figure it is clear that 19CS1-1 (brown curve) grown faster than 19CS9-1.
RESPONSE: We appreciate your observation regarding the growth rates of the isolates in the genus. We agree with your assessment that in the Figure 1, the brown curve representing 19CS1-1 in the genus Caballeronia appears to have a faster growth rate than the blue curve representing 19CS9-1 in the genus Paraburkholderia. However, we would like to choose one representative strain in the three different genera, i.e., 19CS4-2, 19CS9-1, and 19C6. To avoid confusion, we have added growth rate data and rewritten the sentences with a new Figure 1 and Figure 2 as follows:
“The growth of the isolates (19CS4-2, 19CS9-1, 19CS1-1, 19CS2-2, 19CS8, and 19C6) were assessed in the presence of 3-CBA as the carbon source by monitoring turbidity at 600 nm until they reached the stationary phase. Notably, three isolates, 19CS4-2, 19CS9-1, and 19CS1-1, displayed significantly faster growth compared with that of the remaining three isolates and reference strain NK8 (Figure 1). The data are represented as the mean ± standard deviation of three independent experiments.
The growth kinetics were calculated using Monod kinetic model by plotting the graph ln (OD600) vs. incubation time (Figure S2). In the Figure S2, the slope of the curve shows the maximum specific growth rate of the isolates (Eq. 2). The exponential growth rate and doubling time comparison among of the isolates and the reference strain Caballeronia sp. NK8 is shown in Table 3. The maximum specific growth rates and doubling times were varied among the isolates and the reference strain. The 19CS4-2 and 19CS1-1 strains had the highest maximum specific growth rate (µmax) of 0.30 h-1 and 0.29 h-1 and the shortest doubling time compared with the remaining four isolates and the reference strain NK8. The maximum specific growth rate of 19CS9-1 was very close to that of the NK8 strain. These six isolates were then classified into three groups based on their genera. The first group consisted of Caballeronia genera (19CS4-2, 19CS2-2, and 19CS1-1); the second group included Paraburkholderia genera (19CS9-1 and 19C8); and the third group included Cupriavidus genera (19C6). One representative strain from each group (genus Caballeronia strain 19CS4-2, genus Paraburkholderia 19CS9-1, and Cupriavidus 19C6) was selected for further evaluation of their degradation ability and genetic analysis based on their faster growth than the other strains. ” (Lines 215-235 in the revised manuscript)
197: ,, These differences may be due to different metabolic mechanisms being used in the different isolates“. What are these mechanisms? If the author chose not to make a separate part of the discussion, the results must still be discussed in more depth and compared with the results of other researchers.
Figure 2. It is strange that when the OD600 is under 0,1 (almost 0), the concentration of 3-CB decreases by 1 mM (strain 19C6 – pink curve). What is the reason? It must be explained.
RESPONSE: Thank you for your insightful comments and suggestions regarding the metabolic mechanisms underlying the observed differences in growth rates among the isolates in our study. we have added the degradation rate and compared with each other. We also add the HPLC and MS analysis to find out the metabolites present in the isolates for 3-CBA degradation. We have rewritten the sentences with a new figure 3 as follows:
HPLC analysis was performed on the three representative strains based on their growth curves at each time point. The samples were collected at 8, 16, 20, 24, and 28 hours after the start of the incubation period. In Figure S3, two HPLC chromatogram results (after 8 hours and complete reduction of 3-CBA) are shown to illustrate the changes in 3-CBA concentration over time. The analysis showed that 19CS4-2 and 19CS9-1 reduced 5 mM 3-CBA levels to below the detection limit within 20 hours of cultivation, whereas 19C6 required 28 hours to achieve the same level of degradation (Figure 2). The data are represented as the mean ± standard deviation of three independent experiments. The degradation rate of the isolates 19CS4-2, 19CS9-1, and 19C6 were 0.29 mM h-1, 0.23 mM h-1, and 0.10 mM h-1, respectively. 19CS4-2 had a higher degradation rate than the other isolates. The 19C6 isolate was able to degrade 3-CBA with a low growth rate. The low growth rate in 19C6 might be attributable to the presence of a toxic intermediate during the degradation of 3-CBA.The accumulation of these toxic compounds can hinder bacterial growth and metabolic activity [21]. The metabolites were then investigated by MS analysis.
HPLC analysis revealed the presence of two metabolites in all isolates grown on 3-CBA (Figure 3). The two peaks of the isolates were collected 40 hours after the start of the incubation period. The mass spectra of the isolates showed deprotonated molecular ions at m/z=157.02 [M-H]− and m/z=174.99 [M-H]−. These ions correspond to the molecular ions [M−] at m/z=158.02 and m/z=175.99, which are the masses of maleylacetate and chloro-cis,cis-muconate, respectively (Figure 3). The metabolites were found in all isolates, suggesting that they might be involved in the 3-CBA degradation pathway. Although the metabolites were found in all strains, the growth characteristics of strain 19C6 were notably different to those of 19CS4-2 and 19CS9-1. Further genomic analyses were conducted to gain a more detailed understanding of the genetic mechanisms underlying the different degradation abilities of these isolates.” (Lines 248-275 in the revised manuscript)
We have also discussed the reason why 19C6 could degrade 3-CBA with low growth rate in the genomic analysis section as follows, “Two degradation product including maleylacetate and chloro-cis,cis,-muconate, were found in the all isolates, which are the intermediates of the chlorocatechol ortho-cleavage pathway. The slow growth of the 19C6 might happen due to lack of the tfdCD genes involved in degradation of chlorocatechol and chloro-cis,cis,-muconate which are intermediates in the degradation pathway of 3-CB. These products might inhibit the growth of the bacteria[21]. Consequently, even at low OD values, strain 19C6 is capable of reducing the concentration of 3-CBA, possibly owing to the presence of cbe genes responsible for the initial steps of 3-CBA degradation.” (Lines 338-345 in the revised manuscript)
287-288:,, These differences may be due to different metabolic mechanisms being used in the other isolates.“ If the author chose not to make a separate part of the discussion, the results must still be discussed in more depth and compared with the results of other researchers.
301-302:,, These findings demonstrate the metabolic versatility of the isolates and highlight the importance of the further investigation to fully understand their degradation pathways.“ If the author chose not to make a separate part of the discussion, the results must still be discussed in more depth and compared with the results of other researchers.
RESPONSE: Thank you for your insightful comments and suggestions. We have revised the manuscript and add more information. We have discussed in details and the findings with the results of the other researchers as follows:
“In most bacteria 4-HB is metabilzed by 4-HB-3-monooxygenase, resulting in the formation of PC (the PC pathway) (Figure 4b), which serves as a central intermediate [58,59]. 4-HB-3-monooxygenase is encoded by the pobA gene in Pseudomonas strains [60,61].” (Lines 362-364 in the revised manuscript)
“A variety of diverse pca gene have been found in bacteria to have organizations for the PC degradation [58,59,62].” (Lines 367-368 in the revised manuscript)
“These findings suggest 19C6 possesses putative genes involved in 3-HB metabolism via GS. The identification of the GS catabolic pathway is known to be present only in a limited number of members of the Burkholderiaceae family [58]. The identification of the GS catabolic pathway in isolate 19C6 highlights its unique metabolic capabilities and distinguishes it from the other two experimental isolates (19CS4-2 and 19CS9-1).” (Lines 390-395 in the revised manuscript)
“The growth studies indicated that 4-HB degradation via PC is present in all three strains. The studies also indicated that a 3-HB degradation via GS is present in 19C6, which has also been observed in other bacteria, including K. pneumoniae M5a1 [63], Corynebacterium glutamicum RES167 [64], and Rhodococcus sp. strain NCIMB 12038, in which GS has been described as the main ring-cleavage pathway for 3-HB assimilation [65].” (Lines 420-425 in the revised manuscript)
“Although the experimental isolates, including 19CS4-2 and 19CS9-1, lacked specific 3-HB degradation genes, they were still able to grow on 3-HB. This suggests that these strains might employ alternative or unidentified pathways for 3-HB conversion. The similarities of the results of our studies to those of similar studies using C. necator NH9 and Caballeronia sp. NK8 underscores the importance of investigating the metabolic versatility and diversity within bacterial species. The results also show that different strains within the same bacterial genus may exhibit distinct metabolic capabilities and employ different metabolic pathways for the degradation of specific compounds.
The genome-wide comparisons of the experimental isolates with the reference strains revealed significant differences between them. The observed differences in the genomes of the isolates compared with those of the reference strains open up new possibilities for exploration. By studying these differences, it is possible to gain a deeper understanding of the metabolic versatility of the isolates. and these findings highlight the importance of further research to fully understand the degradation pathways of these organisms.” (Lines 431-445 in the revised manuscript)
Reviewer 4 Report
Comments to the manuscript microorganisms-2413234: Isolation and genomic analysis of 3-chlorobenzoate-degrading bacteria from soil
The present manuscript deals with the isolation of new 3-chlorobenzoate degrading strains, their degradation pathways and corresponding gene sequences. Argumentation is fine so far, but final proofs for degradation as supposed are not presented, i.e. no detection of intermediates, no quantification of chloride…These data would dramatically improve the quality of this manuscript. Language is fine. The topic fits to the scope of the journal. Before publication, the following aspects should be considered:
· Line 21-22: This sentence is of general relevance. Therefore, not relevant for the abstract.
· Line 44-46: Relevant biological treatment technologies, i.e. bioscrubber, biotricklingfilter and biofilter, should be named and examples of degradation of these halogenated compounds, better halogenated aromatic compounds (not for bioscrubbers), should be listed. Adequate examples might be:
o Bioscrubber:
§ https://dx.doi.org/10.1002/jctb.2216 TCE
§ https://doi.org/10.1016/j.ces.2007.06.040 1,2-DCE and fluorobenzene
§ https://doi.org/10.1016/j.jhazmat.2013.11.013 PCE, TCE
o Biotrickling filter
§ https://dx.doi.org/10.1016/j.scitotenv.2018.05.278 o-chlorotoluene
§ https://doi.org/10.1080/26395940.2022.2151516 m-dichlorobenzene
o Biofilter
§ https://doi.org/10.1016/j.chemosphere.2020.126358 chlorobenzene
§ https://doi.org/10.1016/j.biteb.2020.100387 TCE
· Line 48: Most probably 3-chlorobenzoate is not used in agriculture directly, but in formation of fertilizers or similar. Please clarify and show more concisely.
· Line 51: Abbreviation CB is mainly used for chlorobenzene. Maybe the abbreviation should be avoided.
· Line 66-67: Considering uncommon degradation pathways it perfectly fits to consider special degradation pathways of 3-substituted catechols.
o Meta cleavage of 3-chlorocatechol:
§ 10.1128/JB.181.4.1309-1318.1999
§ DOI: 10.1007/s00203-004-0681-5
o Chorocatechol dioxygenase with extremely width substrate pattern: Mineralisation of 3-methylcatechol by ortho pathway.
§ https://doi.org/10.1111/1751-7915.12147
o Meta cleavage of substituted chlorocatechols (despite risk of toxic intermediates):
§ DOI: 10.1007/s00253-011-3543-5
· Chapter results: Is it right, that all tests were made with BSM media, containing lactic acid as potential co-subtrate? If yes, it should be clearly pointed out that 3-chlorobenzoate degradation is done in presence of an additional carbon source. If available, it would be very good to see a growth curve of the strains based on sole mineral medium (with addition of 3-CB).
· Line 194-196: Another explanation are different levels of ‘fitness’ of bacteria and their state of induced enzymes. Therefore, it might be more applicable to compare maximum transformation rates). In your case this aspect seems to be not relevant, but maybe it should be mentioned in the explanation.
· Line 207ff: Did you isolate the plasmid and sequenced it? Maybe, relevant enzymes for degradation are encoded on the plasmid, which might explain why strains without plasmid cannot growth as fast as strains with plasmid.
· Line 231: Please change to dihydrodiol derivative as there are functional groups.
· Line 232: Please add the aspect of formation of CO2 via decarboxylaction during rearomatisation of the ring.
· Line 244: Carboxymethylbutenolide is commonly known as dienlactone.
· Figure 3a: It would be very helpful to present reaction products getting out of the pathway, i.e. CO2 in step 2, chloride in step 4 …
· Figure 3a: 3-oxoadipate shows the wrong formula; double bonding is eliminated.
· Figure 3b: 3-oxoadipate-enol-lactone shows the wrong formular, CO2 is eliminated. This underlines, why presentation of reaction products is really helpful.
· Figure 3b: 2nd reaction of 3-HB reaction product is wrong, because using a 1,2-dioxygenase would firstly form a C7-dicarboxy acid. Hence, OH substituent and C2-group are lost.
· Beside transformation kinetics of 3-chlorobenzoate, the presentation of chloride formation and balancing is an essential aspect to underline mineralization of 3-CB. Currently, according to gene sequences a hypothetical pathway is impressively described, but no intermediates (with exception of gentisate or hydroxybenzoate degradation – also unknown whether degradation occurs without lag-phase) are detected or chloride formation is presented. This are core data for a clear proof of mineralization. Growth kinetics are comparable weak proofs.
* what is the clear benefit you this work (scientific goals, application, remediation...). Please point out this aspect more clearly.
.
Author Response
Responses to the comments of Reviewer 4:
Comments to the manuscript microorganisms-2413234: Isolation and genomic analysis of 3-chlorobenzoate-degrading bacteria from soil
The present manuscript deals with the isolation of new 3-chlorobenzoate degrading strains, their degradation pathways and corresponding gene sequences. Argumentation is fine so far, but final proofs for degradation as supposed are not presented, i.e. no detection of intermediates, no quantification of chloride…These data would dramatically improve the quality of this manuscript. Language is fine. The topic fits to the scope of the journal. Before publication, the following aspects should be considered:
RESPONSE: Thank you for your valuable feedback on the manuscript. We appreciate your positive comments regarding the argumentation and language used in the study. In response to your suggestions, we add the detection of intermediates data. We will think about the quantification od chloride during the degradation process in future research. We have revised the introduction part according to your suggestion. We hope that you find the revised version satisfactory and suitable for the publication.
Line 21-22: This sentence is of general relevance. Therefore, not relevant for the abstract.
RESPONSE: We removed the sentence and added new sentence “The compound 3-chlorobenzoate (3-CBA) is a hazardous industrial waste product that can be harmful to human health and the environment.” (Lines 22-23 in the revised manuscript)
Line 44-46: Relevant biological treatment technologies, i.e. bioscrubber, biotricklingfilter and biofilter, should be named and examples of degradation of these halogenated compounds, better halogenated aromatic compounds (not for bioscrubbers), should be listed. Adequate examples might be:
Bioscrubber: https://dx.doi.org/10.1002/jctb.2216 TCE, https://doi.org/10.1016/j.ces.2007.06.040 1,2-DCE and fluorobenzene, https://doi.org/10.1016/j.jhazmat.2013.11.013 PCE, TCE
Biotrickling filter: https://dx.doi.org/10.1016/j.scitotenv.2018.05.278 o-chlorotoluene, https://doi.org/10.1080/26395940.2022.2151516 m-dichlorobenzene
Biofilter: https://doi.org/10.1016/j.chemosphere.2020.126358 chlorobenzene, https://doi.org/10.1016/j.biteb.2020.100387 TCE
RESPONSE: Thank you very much for sharing the reference and the suggestion. We have added the information as follows “Several biological treatment technologies have shown promise in the removal of halogenated compounds. These technologies include bioscrubbers (e.g.,trichloroethylene(TCE), 1,2-dichloroethylene (1,2-DCE)) [5–7] , biotrickling filters (e.g., o-chlorotoluene and m-dichlorobenzene)[8,9], biofilters (e.g., chlorobenzene and TCE) [10,11] and biodegradation.” (Lines 45-50 in the revised manuscript)
Line 48: Most probably 3-chlorobenzoate is not used in agriculture directly, but in formation of fertilizers or similar. Please clarify and show more concisely.
RESPONSE: Thank you very much for the information. We rewritten this as follows “3-Chlorobenzoate (3-CBA) is a halogenated compound used as a raw material ffor the production of fertilizers and industry for several applications.” (Lines 55-56 in the revised manuscript)
Line 51: Abbreviation CB is mainly used for chlorobenzene. Maybe the abbreviation should be avoided.
RESPONSE: We have changed the abbreviation CB to CBA
Line 66-67: Considering uncommon degradation pathways it perfectly fits to consider special degradation pathways of 3-substituted catechols.
Meta cleavage of 3-chlorocatechol:10.1128/JB.181.4.1309-1318.1999, DOI: 10.1007/s00203-004-0681-5
Chorocatechol dioxygenase with extremely width substrate pattern: Mineralisation of 3-methylcatechol by ortho pathway. https://doi.org/10.1111/1751-7915.12147
Meta cleavage of substituted chlorocatechols (despite risk of toxic intermediates): DOI: 10.1007/s00253-011-3543-5
RESPONSE: Thank you very much for sharing the reference and the suggestion. We have added the information as follows: “Only a few bacteria can use the meta-cleavage pathway to degrade CC [24–26]. CC can also be used as important central intermediate in the degradation of various chloroaromatic compounds [27,28]. CC dioxygenase an important enzyme for CC degradation, and this combined with its involvement in the ortho pathway, plays a crucial role in the microbial degradation of chloroaromatic compounds [21,29].” (Lines 68-75 in the revised manuscript)
Chapter results: Is it right, that all tests were made with BSM media, containing lactic acid as potential co-subtrate? If yes, it should be clearly pointed out that 3-chlorobenzoate degradation is done in presence of an additional carbon source. If available, it would be very good to see a growth curve of the strains based on sole mineral medium (with addition of 3-CB).
RESPONSE: Thank you very much for your comment. We did not add lactic acid as a co-substrate. We used BSM media supplemented with only 3-CBA as carbon source.
Line 194-196: Another explanation are different levels of ‘fitness’ of bacteria and their state of induced enzymes. Therefore, it might be more applicable to compare maximum transformation rates). In your case this aspect seems to be not relevant, but maybe it should be mentioned in the explanation.
RESPONSE: Thank you very much for the comment. We explained the degradation rate as follows: “The degradation rate of the isolates 19CS4-2, 19CS9-1, and 19C6 were 0.29 mM h-1, 0.23 mM h-1, and 0.10 mM h-1, respectively” in the revised manuscript (Line 256-257)
Line 207ff: Did you isolate the plasmid and sequenced it? Maybe, relevant enzymes for degradation are encoded on the plasmid, which might explain why strains without plasmid cannot growth as fast as strains with plasmid.
RESPONSE: In our study, we did not specifically isolate and sequence the plasmids from the strains. Based on whole genomic analysis, we found the putative genes contig present on chromosome. But there is one possibility that 19C6 had no tfdCD genes involved in degradation of chlorocatechol and chloro-cis,cis,-muconate which are intermediates in the degradation pathway of 3-CB. These intermediates might inhibit the growth of the bacteria. In contrast, strains 19CS4-2 and 19CS9-1 with plasmid had tfd gene cluster for chlorocatechol degradation. We have also added the discussion it as follows “The slow growth of the 19C6 might happen due to lack of the tfdCD genes involved in degradation of chlorocatechol and chloro-cis,cis,-muconate which are intermediates in the degradation pathway of 3-CB. These products might inhibit the growth of the bacteria[21]. Consequently, even at low OD values, strain 19C6 is capable of reducing the concentration of 3-CBA, possibly owing to the presence of cbe genes responsible for the initial steps of 3-CBA degradation.” . (Lines 338-343 in the revised manuscript)
Line 231: Please change to dihydrodiol derivative as there are functional groups.
RESPONSE: We have changed dihydrodiol to dihydrodiol derivative. (Lines 320 in the revised manuscript)
Line 232: Please add the aspect of formation of CO2 via decarboxylaction during rearomatisation of the ring.
RESPONSE: We have added the information. (Lines 323 in the revised manuscript)
Line 244: Carboxymethylbutenolide is commonly known as dienelactone.
RESPONSE: We have changed Carboxymethylbutenolide to dienelactone. (Lines 336 in the revised manuscript)
Figure 3a: It would be very helpful to present reaction products getting out of the pathway, i.e. CO2 in step 2, chloride in step 4 …
Figure 3a: 3-oxoadipate shows the wrong formula; double bonding is eliminated.
Figure 3b: 3-oxoadipate-enol-lactone shows the wrong formular, CO2 is eliminated. This underlines, why presentation of reaction products is really helpful.
Figure 3b: 2nd reaction of 3-HB reaction product is wrong, because using a 1,2-dioxygenase would firstly form a C7-dicarboxy acid. Hence, OH substituent and C2-group are lost.
RESPONSE: We have corrected these points (Figure 4 in the revised manuscript)
Beside transformation kinetics of 3-chlorobenzoate, the presentation of chloride formation and balancing is an essential aspect to underline mineralization of 3-CB. Currently, according to gene sequences a hypothetical pathway is impressively described, but no intermediates (with exception of gentisate or hydroxybenzoate degradation – also unknown whether degradation occurs without lag-phase) are detected or chloride formation is presented. This are core data for a clear proof of mineralization. Growth kinetics are comparable weak proofs.
RESPONSE: Thank you very much for your valuable comment. We have added the result of intermediates identification data in figure 3 in the revised manuscript .We found two intermediates chloro-cis-cis muconate and maleylacetate which are involved in the 3-CBA degradation pathway. These finding demonstrates the 3-CBA degradation pathway by the obtained isolates.
* what is the clear benefit you this work (scientific goals, application, remediation...). Please point out this aspect more clearly.
RESPONSE: Thank you very much for the suggestion. We have added the benefit of our study as follow: “The genome-wide comparisons of the experimental isolates with the reference strains revealed significant differences between them. The observed differences in the genomes of the isolates compared with those of the reference strains open up new possibilities for exploration. By studying these differences, it is possible to gain a deeper understanding of the metabolic versatility of the isolates. and these findings highlight the importance of further research to fully understand the degradation pathways of these organisms.” (Lines 440-445 in the revised manuscript)
Round 2
Reviewer 2 Report
There are still some minor errors and omissions in the manuscript that need to be corrected:
the units corresponding to the numbers in Table are missing
line 364 the authors still do not indicate what leads them to propose the presence of plasmids in some of the strains
line 391 putative genes ... were found
line 450 which probably are involved ...
line 464 why use dienelactone here and trans-4-carboxymethylene but-2-en-4olide in Fig. 4? I was highly distracted by this.
the persistence of some bad grammar errors at this point is annoying. Instead of getting the paper edited by a "commercial service", it may have been better if the authors spent another look at it themselves.
Author Response
There are still some minor errors and omissions in the manuscript that need to be corrected: the units corresponding to the numbers in Table are missing
RESPONSE: Thank you very much for your valuable feedback on our manuscript. We have corrected the minor errors and omissions in the manuscript. We also have added the units in Table 3.
line 364 the authors still do not indicate what leads them to propose the presence of plasmids in some of the strains
RESPONSE: Thank you very much for the comment. We have rewritten it as follows: “From the comparative analysis of annotation of whole genome sequences of each starin, two isolates 19CS4-2 and 19CS9-1 were predicted to contain plasmids, whereas isolate 19C6 was not. The presence or absence of plasmids in these isolates may have significant implications for their genetic diversity, adaptability, and potential interactions within their respective ecological niches.” (Lines 290-294 in the revised manuscript)
line 391 putative genes ... were found
RESPONSE: Corrected. (Lines 316 and 323 in the revised manuscript)
line 464 why use dienelactone here and trans-4-carboxymethylene but-2-en-4olide in Fig. 4? I was highly distracted by this.
RESPONSE: Thank you very much for the comment. We have modified the Figure 4.
Comments on the Quality of English Language
the persistence of some bad grammar errors at this point is annoying. Instead of getting the paper edited by a "commercial service", it may have been better if the authors spent another look at it themselves.
RESPONSE: According to the reviewer’s comment, we have carefully checked the grammar errors throughout the manuscript by ourselves.
Reviewer 4 Report
Dear authors,
all comments were adressed. Good job.
Best regards
Author Response
Thank you for your positive feedback on our research. We greatly appreciate your assessment.